# Melatonin Supplementation during *In Vitro* Maturation of Porcine Oocytes Alleviates Oxidative Stress and Endoplasmic Reticulum Stress Induced by Imidacloprid Exposure

**DOI:** 10.3390/ani13162596

**Published:** 2023-08-11

**Authors:** Jing Wang, Xin-Qin Wang, Rong-Ping Liu, Ying-Hua Li, Xue-Rui Yao, Nam-Hyung Kim, Yong-Nan Xu

**Affiliations:** 1Guangdong Provincial Key Laboratory of Large Animal Models for Biomedicine, School of Biotechnology and Health Sciences, Wuyi University, Jiangmen 529020, Chinayhli2225@163.com (Y.-H.L.);; 2College of Agriculture, Yanbian University, Yanji 133002, China

**Keywords:** imidacloprid, melatonin, porcine oocytes, oxidative stress, endoplasmic reticulum stress, apoptosis

## Abstract

**Simple Summary:**

The residue of chemicals can affect the maturation of livestock oocytes, thereby affecting the quality of embryos. Research has shown that imidacloprid, a systemic neonicotinoid insecticide widely used in agriculture, can damage the reproductive system of mammals, and it is unclear whether imidacloprid will affect oocyte maturation and how to reduce its toxicity. The results of this study show that imidacloprid can affect oocyte maturation, and melatonin supplementation can restore oocyte maturation by alleviating oxidative stress and endoplasmic reticulum stress, and offset the toxic effect of imidacloprid on pig oocytes, which indicates that Melatonin may be a promising drug to improve the quality of oocytes exposed to imidacloprid in animals.

**Abstract:**

Imidacloprid (IMI) is an endogenous neonicotinoid insecticide widely used in agriculture and has attracted researchers’ attention because of its risks to the environment and human health. Melatonin (MT) is an antioxidant hormone produced by the pineal gland of the brain. Studies have shown that it has a variety of physiological functions and plays a crucial role in the development of animal germ cells and embryos. The potential protective effects of MT against oocyte damage caused by neonicotinoid pesticide toxicity remain unclear. In this study, we report the toxicity of IMI against, and its effects on the quality of, porcine oocytes and the protective effect of MT on IMI-exposed oocytes. The results show that IMI exposure adversely affected oocyte maturation, while MT supplementation ameliorated its toxic effects. Specifically, IMI exposure increased oxidative stress (OS), endoplasmic reticulum stress (ERS), and apoptosis, which may affect polar body expulsion rates and blastocyst formation. Also, IMI exposure reduced oocyte cleavage rates and the number of cells in blastocysts. However, all of these toxic effects can be restored after a melatonin supplementation treatment. In conclusion, these results suggest that melatonin has a protective effect on IMI-induced defects during porcine oocyte maturation.

## 1. Introduction

*In vitro* maturation (IVM) of oocytes refers to the period in which the arrested oocytes resume meiosis and enter metaphase II (MII). It is the first and most critical stage in the process of *in vitro* embryo production [1]. The process of oocyte maturation is particularly susceptible to environmental pollutants and chemical substances [2,3]. This can lead to a decrease in oocyte quality, which can lead to female infertility, miscarriage, and congenital disorders in the fetus [4].

Imidacloprid (IMI) is a nitromethyl compound, and because of its effectiveness against insects, it is the world’s best-selling neonicotinoid [5]. Widespread use of IMI in agriculture leads to its persistence in soil, water, and plants. Research indicates that IMI can not only damage the central nervous system of insects, causing damage such as oxidative stress and death [6,7], but also damage the nervous, immune, and reproductive systems of mammals and poultry [8,9]. IMI administration can lead to oxidative stress in pigeons [10], and exposure to IMI can induce oxidative stress, endoplasmic reticulum (ER) stress, inflammation, and apoptosis in mouse liver [11]. However, other researchers have reported that oral melatonin protects honeybees from imidacloprid-induced oxidative stress [12], however, the effects of IMI on porcine oocytes and its mechanism are still unclear.

Melatonin (5-methoxy-*N*-acetyltryptamine, MT) is an amine hormone mainly secreted by the pineal gland at night [13]. Melatonin can inhibit oxidative stress and early apoptosis of germ cells and improve sperm viability in assisted reproductive therapy [14,15]. It can effectively maintain the healthy morphology of oocytes, delay the decline of mitochondrial membrane potential of senescent oocytes, induce oocyte maturation, enhance the developmental ability of porcine nucleus, and promote embryonic development [16,17,18]. Compared with other antioxidants, melatonin has the advantages of fast metabolism and less damage to oocytes [19,20].

The purpose of this study was to investigate the effect of IMI exposure on porcine oocyte maturation and its mechanism of action, and to explore whether melatonin can alleviate IMI-induced oxidative stress, endoplasmic reticulum stress, apoptosis, and reductions in embryonic development rates following parthenogenesis activation.

## 2. Materials and Methods

All chemicals and reagents were purchased from Sigma-Aldrich (St. Louis, MO, USA) unless stated otherwise.

### 2.1. Porcine Oocyte Collection and IVM

Porcine ovaries were taken from the local slaughterhouse (Jiang Xin meat factory, Jiangmen, China) and transported to the laboratory in sterile 0.9% saline at 30–35 °C within 2 h. Cumulus–oocyte complexes (COCs) were extracted from 3 to 6 mm follicles using a 10 mL syringe and an 18-gauge needle. Oocytes with at least three layers of cumulus cells were washed twice with Tyrode’s lactate HEPES (TL-HEPES), then three times with IVM medium (TCM-199 maturation medium (Thermo Fisher Scientific, #11150-059, Waltham, MA, USA) supplemented with 10% porcine follicular fluid, 0.57 mM L-cysteine, 20 ng/mL epidermal growth factor, 1% Penicillin–Streptomycin (Thermo Fisher Scientific, #15140122, Waltham, MA, USA), 0.2 mM sodium pyruvate, 10 IU/mL follicle-stimulating hormone [Ningbo No. 2 Hormone Factory, Ningbo, China), and 10 IU/mL luteinizing hormone (Ningbo No. 2 Hormone Factory, Ningbo, China). Washed oocytes were then selected for further experiments. Approximately 100 COCs were transferred to 500 µL of mineral oil-covered IVM medium and the oocytes were cultured in an incubator set to 5% CO_2_, 100% humidity, and 38.5 °C for 42–44 h. After maturation, cultures were observed then the cumulus cells were blown in 0.1% hyaluronidase (*w*/*v*) for 2–3 min before the experiment. Mature oocytes containing the first polar body were selected for parthenogenetic activation.

### 2.2. Imidacloprid and Melatonin Treatments

Treatments were administered during the maturation process described in Section 2.1. Imidacloprid (Solarbio, #SI8240, Beijing, China) solution was added to the TCM-199 maturation medium to final concentrations of 0 μM, 250 μM, 500 μM, and 1000 μM. For testing IMI and MT together, one IMI concentration, 500 μM, was used. The optimal concentration of MT has previously been reported as 1 × 10^−9^ M [21,22,23], which was used throughout this study. It was added to the maturation medium as soon as the COCs started to be cultured.

### 2.3. Oocyte Parthenogenetic Activation and In Vitro Culturing

Denuded oocytes with polar bodies and homogeneous cytoplasm were selected and gradually balanced in the activation solution containing 300 mM mannitol, 0.05 mM CaCl_2_, 0.1 mM MgSO_4_, 0.01% polyvinyl alcohol (PVA, *w*/*v*), and 0.5 mM HEPES. Then, these oocytes were activated by two 120 V DC pulses for 60 μs. The putatively activated oocytes were then cultured in porcine zygote medium-5 (PZM-5) containing 4 mg/mL BSA and 7.5 μg/mL cytochalasin B for 3 h to suppress the extrusion of the pseudo-second polar body.

Then, the oocytes were thoroughly washed and cultured in four-well plates in PZM-5 medium covered with mineral oil, the oocytes were cultured in an atmosphere of 38.5 °C, 100% humidity, and 5% CO_2_ for 7 days without changing the medium. The cleavage rate and blastocyst rate on day 2 and day 7 were analyzed under a stereomicroscope.

### 2.4. Intracellular ROS and GSH Level Assays

The intracellular ROS and GSH levels after oocyte maturation (non-activated) were detected by ROS detection kit (Thermo Fisher Scientific, #C400) and GSH level detection kit (Thermo Fisher Scientific, #C12881), respectively. To determine intracellular ROS levels, oocytes were incubated in 0.1% PBS-PVA medium containing 10 μM 2′,7′ dichlorodihydrofluorescein diacetate (H2DCFDA) for 30 min. To determine intracellular GSH levels, oocytes were incubated in 0.1% PBS-PVA medium containing 10 μM 4-chloromethyl-6,8-difluoro-7-hydroxycoumarin (CMF2HC) for 30 min. After washing thrice with PBS-PVA, fluorescence microscopy (Eclipse Ti2; Nikon, Tokyo, Japan) and ImageJ software (NIH, Bethesda, MD, USA, https://ij.imjoy.io/) were used to analyze the fluorescence intensities.

### 2.5. Measurement of Cathepsin B Activity

Cathepsin B activity was detected using a Magic Red Cathepsin B Assay Kit (#938, Immuno Chemistry Technologies, Bloomington, MN, USA) according to the manufacturer’s instructions. After 44 h of culture, cumulus cell-removed oocytes were placed in 25 μL 0.1% PBS-PVA, 1 μL reaction solution was added, and incubated at 37 °C for 30 min in the dark. Fluorescence signals were captured using fluorescence microscopy and analyzed using Image J software.

### 2.6. Assessment of Blastocyst Total Cell Numbers

In order to determine the total number of cells in blastocysts, blastocysts from parthenogenetic-activated embryos developed to day 7 were collected, fixed in 0.1% PBS-PVA medium containing 3.7% paraformaldehyde for 30 min, then incubated in 0.3% Triton X-100 at room temperature for 30 min. Following incubation, they were treated with 10 μg/mL Hoechst 33,342 and incubated in the dark at 37 °C for 15 min. After that, the stained blastocysts were gently mounted onto glass slides, examined, and photographed with a microscope under fluorescent light. The total number of blastocyst cells were analyzed with ImageJ software.

### 2.7. Relative Gene Expression Measurement Using Real-Time Reverse Transcription Polymerase Chain Reaction (RT-PCR)

Total mRNA was extracted from approximately 100 MII oocytes using a Dynabeads mRNA DIRECT Purification Kit (Invitrogen #61012, Dynal Asa, Oslo, Norway). Next, a Dynabeads mRNA Direct Kit (Invitrogen #18080-051) was used to reverse transcribe the extracted RNA, obtaining cDNA. The qPCR reaction was performed using a CFX Connect Optics Module (Roche/Light Cycler 96) and a Kapa Kit (#KK4600). Each 20 µL qRT-PCR reaction mixture included 8 µL of deionized water, 10 µL of SYBR green, 1 µL of cDNA, and 0.5 µL each of the forward and reverse primers (10 mM). The following settings were used for the thermal cycler: initial denaturation at 95 °C for 3 min, followed by 40 cycles of denaturation at 95 °C for 3 s, annealing at 60 °C for 30 s, and extension at 72 °C for 20 s. The target genes included genes associated with oocyte maturation—*MOS*, *CCNB1*, *GDF9*, and *BMP15*—endoplasmic reticulum stress-related genes—*GRP78*, *IRE1*, *JNK*, *XBP1*, and *CHOP*—oxidative stress-related genes—*CAT*, *SIRT1*, and *SOD1*, and apoptosis related genes—*BAX*, *BCL2*, and *CASPASE-3*. The gene encoding glyceraldehyde-3-phosphate (*GAPDH*) was used as a reference. The primers used to amplify each gene are shown in Table 1. The mRNA quantification data were analyzed using the 2^−ΔΔCT^ method.

### 2.8. Statistical Analysis

The results were expressed as means ± standard deviations (SD). The total number of MⅡ oocytes used (n) and the number of independent repetitions (R) used in each repeat of the experiment are shown in the figures. For each variable, we measured one-way analysis of variance followed by Tukey–Kramer tests were used to test for significant differences and to compare individual means when applicable. All statistical analyses were performed using SPSS version 17.0 (IBM, Chicago, IL, USA). Significant differences are represented by * (*p* < 0.05), ** (*p* < 0.01), and *** (*p* < 0.001).

## 3. Results

### 3.1. MT Alleviates Maturation Rate and Oocyte Quality Reductions in IMI-Exposed Porcine Oocytes

Compared to the control group, the cumulus expansion of COCs in the IMI treatment groups was reduced to varying degrees and, in some cases, cumulus cells exhibited no proliferation after being cultured for 44 h (Figure 1A). In the control group, most of the oocytes were able to extrude the first polar body and reach the stage of MII (76.17 ± 10.71%). However, the proportion of oocytes reaching the MII stage significantly decreased after IMI treatment at higher doses: at 250 μM, 69.66 ± 3.66% still reached the MII stage; at 500 μM, 50.09 ± 5.84% did, which was significantly less than the control group (*p* < 0.01); at 1000 μM, only 28.88 ± 5.97% reached the MII stage, again, significantly less than that of the controls (*p* < 0.001, Figure 1B). This shows that the effects of IMI on oocyte maturation are dose-dependent. IMI treatment at 500 μM was chosen for subsequent experiments.

We investigated whether MT (1 × 10^−9^ M) could alleviate the decrease in oocyte maturation caused by IMI (Figure 1C). The oocyte maturation rate in the IMI + MT group (70.63 ± 5.22%) was not significantly lower than the control group’s (73.66 ± 2.87%), and both were significantly higher than the IMI group (53.08% ± 3.47%, *p* < 0.01). According to these results, MT can effectively alleviate the negative effects of IMI on oocyte maturation. Concomitantly, MT supplementation alleviated gene expression reductions in developmental competency (*BMP15* and *GDF9*), mitogen-activated protein kinase (*MOS*), and maturation-promoting factor (*CCNB1*) in IMI-exposed oocytes but did not affect reductions in *GDF9* (Figure 1D). 

### 3.2. MT Reduces Oxidative Stress and Endoplasmic Reticulum Stress in IMI-Exposed Porcine Oocytes

The ROS and GSH levels in the IMI-exposed oocytes were examined (Figure 2A–D). Our results showed that IMI induced oxidative stress in oocytes: ROS signal was significantly increased (1.54 ± 0.58 vs. 1.00 ± 0.27, *p* < 0.001), and the GSH signal significantly lower (0.60 ± 0.26 vs. 1.00 ± 0.16, *p* < 0.001) in the IMI-exposed group compared with the control group, while in the IMI + MT group, there was a negligible ROS signal change (0.96 ± 0.40 vs. 1.00 ± 0.27) and the change in the GSH signal (0.82 ± 0.16 vs. 1.00 ± 0.27) was insignificant. Antioxidative stress-related genes—*CAT*, *SIRT1*, and *SOD1*—were assessed, and it was found that MT supplementation could significantly alleviate the effects of IMI exposure, thus improving the ability of oocytes to resist oxidative stress.

Endoplasmic reticulum stress can also lead to decreases in oocyte maturation rates. We examined endoplasmic reticulum stress-related genes—*GRP78*, *IRE1*, *JNK*, *XBP1*, and *CHOP*—and found that MT can alleviate the increases in expression of endoplasmic reticulum stress genes associated with IMI exposure (Figure 3B).

### 3.3. MT Reduces Apoptosis in IMI-Exposed Porcine Oocytes

The CB signal was significantly higher in the IMI-exposed group compared with the control group (1.37 ± 0.35 vs. 1.00 ± 0.19, *p* < 0.001), while there was a negligible CB signal change in the IMI + MT group (1.01 ± 0.24 vs. 1.00 ± 0.19, Figure 4A,B). In addition, MT supplementation attenuates the effects of IMI exposure on the expression of apoptosis-related genes (Figure 4C).

### 3.4. MT Promotes Blastocyst Formation in IMI-Exposed Porcine Embryos after Parthenogenetic Activation

We further evaluated whether IMI exposure during the IVM period impaired the developmental competence of parthenogenetically activated porcine embryos. The results show that IMI exposure had a negative effect (Figure 5A). As shown in Figure 5B–D, the cleavage rate (75.97 ± 3.84% vs. 91.48 ± 3.12% on day 2, *p* < 0.05), blastocyst formation rate (21.33 ± 3.70% vs. 38.02 ± 2.68% on day 7, *p* < 0.001), and the total cell numbers (27.03 ± 6.57 vs. 49.71 ± 6.34, *p* < 0.001) of the parthenogenetically activated embryos generated from matured oocytes in the IMI-exposed group were significantly lower than those in the control group. However, when supplemented with MT, the damage caused by IMI exposure was alleviated, and cleavage rates (87.82 ± 2.13%), blastocyst formation rates (34.19 ± 2.91%), and the total cell numbers (46.19 ± 6.16%) were not significantly different from the controls.

## 4. Discussion

Neonicotinoid insecticides are the most widely used synthetic insecticides in the world [24]. Due to their water solubility and long-term persistence in the environment, they have caused serious environmental problems [25]. Insecticides can enter the body through ingestion, inhalation, or skin contact [26,27]. Imidacloprid (IMI) is one of the top used and the most well-known broad-spectrum, systemic, neonicotinoid pesticides used extensively against sucking, boring, and root-feeding insects, representing more than 25% of the world’s pesticide market. In addition, it is also used in many veterinary drugs to treat pet fleas [28]. There are reports that IMI exposure has resulted in sperm toxicity [29], testicular defects, and disturbance of the reproductive system in male rats [30] and altered ovarian morphology in female rats [31]. If sows mistakenly eat the plant sprayed with imidacloprid in the feeding process, whether it will affect the reproductive performance is a question that needs to be discussed. Therefore, this experiment will expose pig oocytes to the environment of imidacloprid for *in vitro* maturation, and use melatonin to alleviate the negative effects of imidacloprid. In this study, we found that IMI at a concentration of 250 μM reduced the polar body excretion rate of oocytes but had no significant effect; 500 μM and 1000 μM treatment significantly reduced the rate of polar body excretion, however, 1000 μM had too much effect on the quality of oocytes, and most oocytes died. This is similar to other research findings, which suggest that the use of high-dose additives in *in vitro* embryo production can have significant harmful effects [32,33]. Therefore, 500 μM is selected as the follow-up experiment. From the experimental results, imidacloprid can affect oocyte maturation, increase oxidative stress, endoplasmic reticulum stress and apoptosis, and affect the early development of oocytes (Figure 6A). In addition, we demonstrated that MT can overcome the decrease in pig oocyte quality caused by exposure to IMI (Figure 6B).

The expansion of cumulus cells and the expulsion of polar bodies are two important indicators of oocyte maturation [34]. Cumulus cells continuously provide nutrients during oocyte growth and maturation [35]. Poor expansion of cumulus cells always leads to a decrease in maturation rate and oocyte quality [36]. Our results showed that IMI exposure significantly decreased the rate of porcine oocyte maturation and cumulus cell expansion, reducing the rate of polar body extrusion, in a dose-dependent manner. In addition, the proto-oncogene serine/threonine kinase (*MOS*), cyclinB1 (*CCNB1*), growth differentiation factor 9 (*GDF9*), and bone morphogenetic protein 15 (*BMP15*) genes are essential for meiosis [37,38,39]. In this study, the mRNA levels of *MOS*, *CCNB1*, *GDF9*, and *BMP15* decreased after IMI treatment, indicating that IMI affects porcine oocyte meiosis. This again suggests that IMI exposure is toxic to oocyte maturation. We found that MT supplementation alleviated these effects too, implying it can restore meiosis and improve oocyte quality. These results are similar to those of previous studies [21,40,41]. Overall, IMI clearly impairs oocyte quality and MT mitigates this damage.

Intracellular ROS and GSH play multiple roles in redox regulation and cell signaling [42]. However, during *in vitro* culture, excessive accumulation of ROS can lead to mitochondrial dysfunction, apoptosis, and meiotic arrest, resulting in impaired embryonic development [43,44,45]. GSH is a major, non-protein sulfhydryl compound in mammalian cells that plays a crucial role in protecting oocytes against oxidative damage [46]. Previous papers have reported that IMI can cause ROS-mediated lipid peroxidation in Caco-2 and HepG2 cells [47]. Furthermore, IMI-induced oxidative stress might be associated with the NF-kappaB/JNK signaling pathway [48] and lead to oxidative stress and DNA damage in earthworms, reducing the expression of *SOD* and *CAT* related antioxidant stress genes [49]. Melatonin, as a free radical scavenger, promotes oocyte maturation and gene expression related to antioxidant pathways [50]. Specifically, supplementation of melatonin during *in vitro* maturation of bovine oocytes can alleviate oxidative stress caused by Juglone, alleviate mitochondrial dysfunction and oxidative stress of mouse oocytes after exposure to Sudan I, and improve oxidative stress and apoptosis of pig oocytes induced by ochratoxin A [16,51,52]. As shown in Figure 2 and Figure 3A, the results of this study showed that IMI exposure to porcine oocytes can significantly increase intracellular ROS levels, reduce GSH levels and the expression of antioxidant genes including catalase (*CAT*), superoxide dismutase 1 (*SOD1*), and silent information regulator 1 (*SIRT1*). We also found that melatonin can effectively alleviate these effects, and thus oxidative stress caused by IMI exposure.

The endoplasmic reticulum (ER) is involved in a variety of cellular functions through its control of protein synthesis, calcium homeostasis, or phospholipid synthesis and plays a key role in oocyte meiotic maturation [53]. Any perturbation in the function of the ER induces the activation of the unfolded protein response (UPR) [54,55]. If misfolded proteins are overloaded or ER stability is not restored, UPR-mediated apoptosis will be triggered [56]. UPR-mediated apoptosis is activated by pro-apoptotic transcription factor C/EBP homologous protein (*CHOP*), apoptosis signal-regulated kinase 1 (*ASK1*)/c-Jun N-terminal kinase (*JNK*) cascade, and *Bax*/*Bcl2*. In addition, oxidative damage or cytotoxicity disrupts ER homeostasis by activating the ER stress UPR during reproduction [57,58]. There are reports that MT inhibits isoflurane-induced endoplasmic reticulum stress and apoptosis [59]. As shown in Figure 3B and Figure 4, our results showed that IMI exposure causes oocyte ERS (specifically, the ERS genes *GRP78*, *IREI*, *JNK*, *XBP1*, *CHOP* were significantly increased) and induces the onset of apoptosis (specifically, the levels of cathepsin B and pro-apoptotic genes *Bax*, *Caspase-3* were significantly increased, and anti-apoptotic *Bcl2* genes were significantly decreased). Therefore, we suggest that IMI disrupted the homeostasis of the ER, which may further damage porcine oocyte maturation and early embryonic development. The addition of melatonin could alleviate endoplasmic reticulum stress and apoptosis and help maintain healthy embryonic development. Our results were similar to previous studies [60,61,62,63], indicating that melatonin can effectively alleviate the damage caused by IMI exposure.

## 5. Conclusions

In summary, our results demonstrate that IMI exposure causes impaired meiosis, induces oxidative stress, and endoplasmic reticulum stress-related apoptosis, which further affects oocyte maturation and embryonic development. Melatonin, due to its antioxidant properties, has a protective effect on IMI toxicity and is a promising pharmaceutical preparation.

## Figures and Tables

**Figure 1 animals-13-02596-f001:**
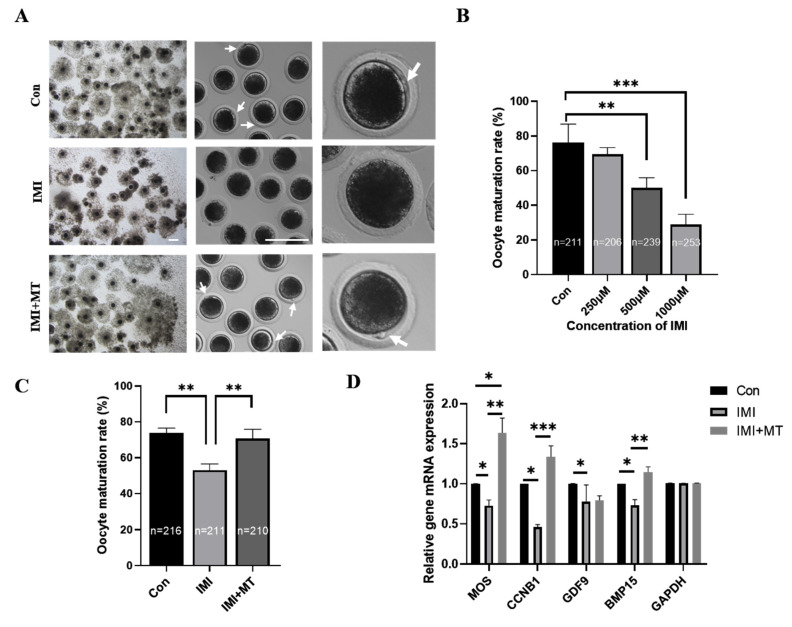
The effects of imidacloprid (IMI) and melatonin (MT) exposure on oocyte maturation and gene expression. (**A**) Oocyte maturation in untreated control media (Con) and after an IMI treatment alone and in conjunction with an MT supplement (IMI + MT). Scale bar = 200 μM; arrows point to polar body. (**B**) The effect of different concentrations of IMI on the polar body extrusion (maturation) rate in porcine oocytes (R = 5; ** *p* < 0.01; *** *p* < 0.001). (**C**) The rate of polar body extrusion after IMI exposure alone and in conjunction with an MT supplement (R = 5; ** *p* < 0.01). (**D**) Relative expression of oocyte competence-related genes after IMI exposure alone and in conjunction with an MT supplement (R = 3; * *p* < 0.05; ** *p* < 0.01; *** *p* < 0.001). The total numbers of oocytes examined from the different groups are indicated on the bars (**C**,**B**).

**Figure 2 animals-13-02596-f002:**
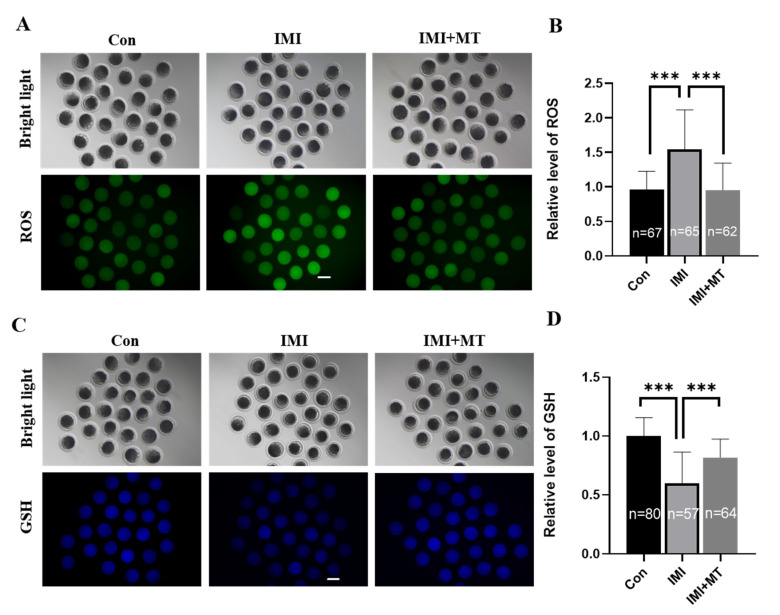
Effects of imidacloprid (IMI) exposure and melatonin (MT) supplementation on intracellular ROS levels and GSH activity in porcine oocytes. (**A**) Representative fluorescence images showing intracellular ROS levels in porcine oocytes in the control (Con), IMI exposure, and IMI exposure in conjunction with MT supplementation (IMI + MT) groups at the end of the IVM period. Scale bar = 200 μM. (**B**) Quantification of relative intracellular ROS levels in porcine oocytes from the control, IMI exposure, and IMI + MT groups. (**C**) Representative fluorescence images of GSH-stained porcine oocytes in the control, IMI exposure, and IMI + MT groups. Scale bar = 200 μM. (**D**) Quantification of relative intracellular GSH levels in porcine oocytes from the control, IMI exposure, and IMI + MT groups. The total numbers of oocytes examined from the different groups are indicated in the bars (R = 3; *** *p* < 0.001).

**Figure 3 animals-13-02596-f003:**
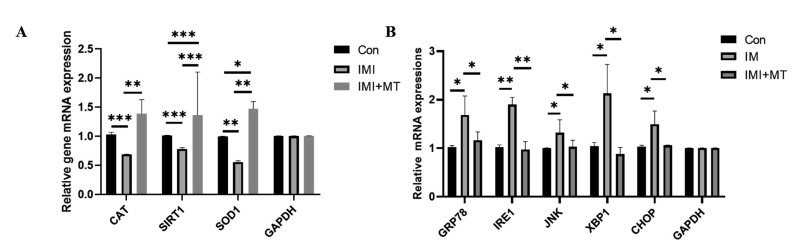
Relative expression of MII stage oocyte genes in untreated control oocytes (Con), imidacloprid (IMI)-exposed oocytes, and oocytes exposed to IMI in conjunction with a melatonin supplement (IMI + MT). (**A**) The mRNA levels of oxidative stress-related genes. (**B**) The mRNA levels of endoplasmic reticulum stress-related genes (R = 3; * *p* < 0.05; ** *p* < 0.01; *** *p* < 0.001).

**Figure 4 animals-13-02596-f004:**
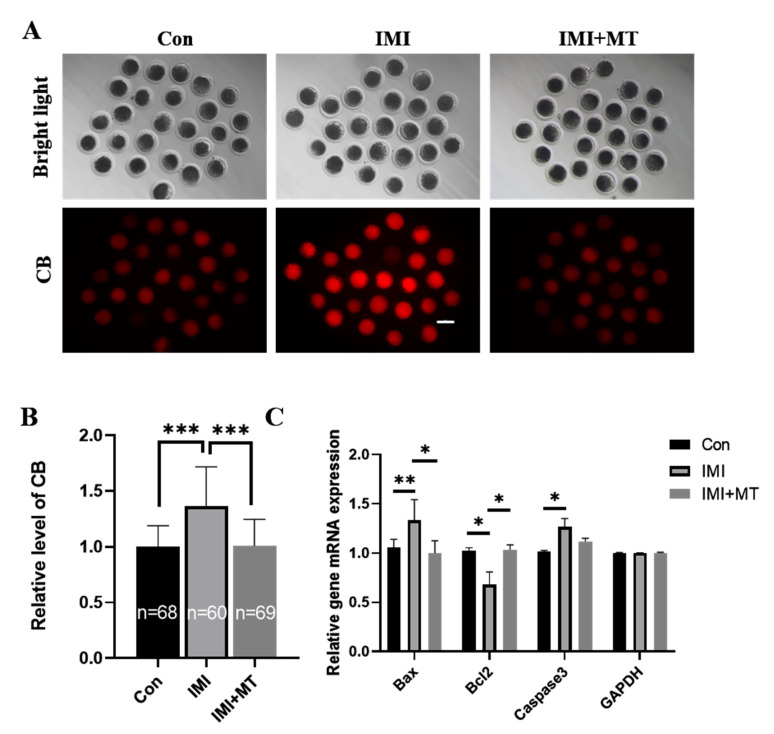
Effects of imidacloprid (IMI) exposure and melatonin (MT) supplementation on intracellular cathepsin B (CB) activity and associated gene expression. (**A**) Representative fluorescence images of CB activity in porcine oocytes of the control (Con), IMI exposure, and IMI exposure in conjunction with MT supplementation (IMI + MT) groups. Scale bar = 200 μM. (**B**) Quantification of relative intracellular CB levels in porcine oocytes from the control, IMI exposure, and IMI + MT groups. The numbers of oocytes examined from the different groups are indicated on the bars (R = 3; *** *p* < 0.001). (**C**) Related gene expression levels (R = 3; * *p* < 0.05; ** *p* < 0.01).

**Figure 5 animals-13-02596-f005:**
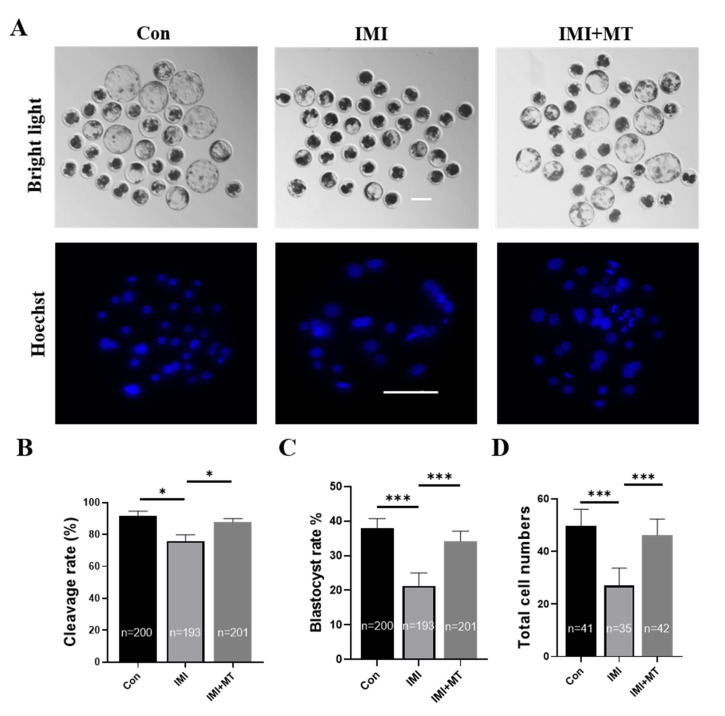
Effects of imidacloprid (IMI) exposure and melatonin (MT) supplementation on *in vitro* development after oocyte activation. (**A**) Representative images of embryo development (**top row**) and Hoechst 33342 staining of blastocysts on day 7 (**bottom row**) in the control (Con), IMI exposure, and IMI exposure in conjunction with MT supplementation (IMI + MT) groups. Scale bar = 200 μM. (**B**) Cleavage rate in the control, IMI exposure, and IMI + MT groups. (**C**) Blastocyst formation rate in the control, IMI exposure, and IMI + MT groups. (**D**) Blastocyst total cell number in the control, IMI exposure, and IMI + MT groups. In all graphs, the numbers of oocytes examined from the different groups are indicated on the bars (R = 5; * *p* < 0.05; *** *p* < 0.001).

**Figure 6 animals-13-02596-f006:**
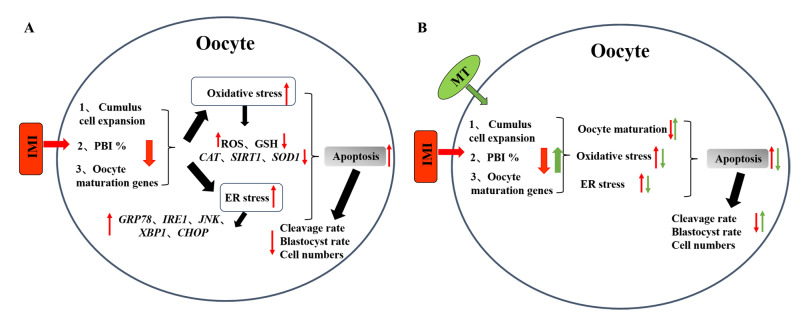
Hypothetical model of imidacloprid IMI and melatonin MT in oocyte maturation. The red arrow represents the negative effects of IMI on oocytes after contact. The green arrow represents the positive effect of MT supplementation on alleviating IMI. (**A**) The effect of IMI exposure on oocytes. IMI contacts oocytes to produce oxidative stress and endoplasmic reticulum stress, which have a negative impact on oocyte maturation and embryonic development potential, and ultimately lead to apoptosis. (**B**) MT alleviates the negative effects of oocytes exposed to IMI. MT can alleviate oxidative stress and endoplasmic reticulum stress, make oocytes mature healthily, and improve embryonic development potential.

**Table 1 animals-13-02596-t001:** Primer sequences (F: forward primer; R: reverse primer), the target gene name, target product size, and the accession number of the entire target gene region.

Genes	Sequences 5′–3′	Product Size (bp)	Accession Number
*GAPDH*	F: TTCCACGGCACAGTCAAG	117	NM_001206359.1
	R: ATACTCAGCACCAGCATCG		
*MOS*	F: GGTGGTGGCCTACAATCTCC	136	NM_001113219.1
	R: TCAGCTTGTAGAGCGCGAAG		
*CCNB1*	F: CCAACTGGTTGGTGTCACTG	195	NM_001170768.1
	R: GCTCTCCGAAGAAAATGCAG		
*BMP15*	F: ATGCTGGAGTTGTACCAGCG	87	NM_001005155.2
	R: CTGAGAGGCCTTGCTCCATT		
*GDF9*	F: CCCCAAAGCCAACAGAAGTCA R: TGATGGAAGGGTTCCTGTCACC	85	NM_001001909.1
*CAT*	F: AACTGTCCCTTCCGTGCTA R: CCTGGGTGACATTATCTTCG	83	XM_021081498.1
*SIRT1*	F: GAGAAGGAAACAATGGGCCG R: ACCAAACAGAAGGTTATCTCGGT	150	NM_001145750.2
*SOD1*	F: CAAAGGATCAAGAGAGGCACG R: CGAGAGGGCGATCACAGAAT	84	NM_001190422.1
*BAX*	F: GCTTCAGGGTTTCATCCAGGATCG R: ACTCGCTCAACTTCTTGGTAGATC	107	XM_003127290.5
*BCL2*	F: GGATAACGGGAGGCTGGGATG R: TTATGGCCCAGATAGGCACC	148	XM_021099593.1
*CASPASE-3*	F: TGTGGGATTGAGACGGACAG R: TTTCGCCAGGAATAGTAACCAGG	116	NM_214131.1
*GRP78*	F: CGGAGGAGGAGGACAAGAAGGAG R: ATATGACGGCGTGATGCGGTTG	143	XM_001927795.7
*IRE1*	F: ACCGTGGTGTCTCAGGATGTGG R: CCAGCCAATGAGCAGGAAGGTG	126	XM_005668695.3
*JNK*	F: CTCGCTACTACAGAGCACCTG R: TTCTCCCATAATGCACCCCAC	85	XM_021073087.1
*XBP1*	F: GGAGTTAAGACAGCGCTTGG R: GAGATGTTCTGGAGGGGTGA	142	NM_001271738.1
*CHOP*	F: TCTGGCTTGGCTGACTGAGGAG	139	NM_001144845.1
	R: TTTCCGTTTCCTGGGTCTTCTTTGG		

Note: The annealing temperature for all reactions was 60 °C.

## Data Availability

The data presented in this study are available on request from the corresponding author.

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
