# Peer review of "Melatonin Supplementation during In Vitro Maturation of Porcine Oocytes Alleviates Oxidative Stress and Endoplasmic Reticulum Stress Induced by Imidacloprid Exposure"

_animals, 2023, doi:10.3390/ani13162596_

Round 1

Reviewer 1 Report

Animals 2480046

Tittle: Melatonin supplementation during....

General comments:

A systematic experiment was carried out to prove the effect of melatonin (MT) in scavenging the adverse effect of imidacloprid (IMI) as a toxic agent. The experiment was designed in stages, beginning with the effect of IMI concentration on oocyte maturation rate and progressing to confirm the expression of related genes. The method of selecting the lethal dose of IMI was accepted by comparing it to the other dose group. The gene expression result was also compared between the control treatment and the addition of MT. Upstream and downstream gene validators or confirmations are also taken into account when selecting genes for expression analysis. The accumulation of MT was also observed in the ROS and antioxidant levels. The expression of BAX and BCL2 genes confirms the effect of MT on oocyte response. However, only a tiny portion of the IMI effect on ROS generation was discussed in the discussion section, and the impact of MT on ROS scavenging was minor.

Specific comments:

1.       As shown in Figures 1, 3, and 4, the expression of GAPDH as a reference gene must be demonstrated in gene expression study analysis. It could be used as an insert or as a standalone figure. I expected the groups to exhibit the same (not significantly different) expression. This is critical information. If the expression is significantly different, the data cannot be used, the paper must be rejected, or other reference genes must be analyzed.

2.       Line 153. The results are presented as means SD, but there is no information on how many replicates there are for each parameter of data collection. Is it biological replication or technical replication? Please include an explanation.

3.       Figure 6 must be explained, and possibly cited in the body text. It is suggested that this figure be moved to the discussion section and the title be replaced with a hypothetical model of IMI and MT in oocyte maturation. A clear explanation of the red and green arrows is also required. It is preferable to show two diagrams. The first is the effect of IMI on OS and ERS generation, as well as the downstream gene target, which reduces oocyte quality and maturation. The second diagram depicts the action of MT in defending the oocyte from the IMI effect.

4.       Look at lines 299–303. Add the graph's reference result as shown to remind the reader of the outcome.

5.       See lines 314-315, similar to comments 4, and add which result in part.

6.       See line 319...our results were similar to previous studies. Which studies? A reference should be included here.

English was fine, a minor grammar can be updated

Reviewer 2 Report

The authors used porcine oocytes to investigate how the toxicity of imidacloprid (IMI) and the protective effects of melatonin on oocyte maturation manifest in vitro. They found that IMI exposure increased oxidative stress (OS), endoplasmic reticulum stress (ERS), and apoptosis, which reduced oocyte cleavage rates and the number of cells in blastocysts. But all of these toxic effects were restored after a melatonin supplementation treatment. Despite the interesting findings, the following concerns remain.

Major Comments

Authors need to clarify under what circumstances this experiment suppose that IMI, itself could be directly exposed to oocytes either in vivo and in vitro since IMI is known to be metabolized in both plants and mammals including human to produce a variety of metabolites such as desnitro-imidacloprid. Actually, many papers indicate IMI toxicity using in vivo system as indicated in this manuscript references #25, 26, and 27. Similarly, authors need to clarify what circumstances 500 μM or higher IMI could be exposed to oocytes and to discuss.

Reviewer 3 Report

The paper emphasizes melatonin function and role very clear, which is nothing new, but little is said about it for swine oocyte maturation.

The study is written in a simple and objective way, making an easy-to-understand reading. The data is presented in a logical and clear way, with very good illustrations. The results found may contribute to the improvement of this first stage, which is so critical, in the in vitro production of swine embryos.

Some small doubts, could leave better details, if clarified:

- In material and methods (line 71), what is the average age of the slaughtered females?

- Did you use only one kind of medium for the entire maturation period (line 83)? Was there no change in half the time, as suggested by modern protocols? Why?

- How many bilogical replicates or oocytes/embryos were done/used, for each evaluation? Were the collected samples taken together or separately?

- Lines 91-92: Was it thought of using only melatonin as one of the treatments? Why was it not included? Would it be better or worse than the control treatment?

- Was it considered to test other concentrations (less than 500 uM) of IMI, which could be less harmful?

- Line 146: How was the GAPDH gene chosen to be the constitutive?

In Abstract, Introduction and Conclusion, some sentences can be rewritten to be less repetitive.

Round 2

Reviewer 1 Report

Dear author see the respond in the review file.

There is still 2 issues, first about the GAPDH graph (its need to be inserted in the figure of gene expression), and second is Figure 6., tittle per ilustration need to be added.

rgds

-
